# Towards Estimating Transferability using Hard Subsets

## Abstract

As transfer learning techniques are increasingly used to transfer knowledge from the source model to the target task, it becomes important to quantify which source models are suitable for a given target task without performing computationally expensive fine-tuning. In this work, we propose HASTE (HArd Subset TransfErability), a new strategy to estimate the transferability of a source model to a particular target task using only a harder subset of target data. By leveraging the model's internal and output representations, we introduce two techniques – one class-agnostic and another class-specific – to identify harder subsets and show that HASTE can be used with any existing transferability metric to improve their reliability. We further analyze the relation between HASTE and the optimal average log-likelihood as well as negative conditional entropy and empirically validate our theoretical bounds. Our experimental results across multiple source model architectures, target datasets, and transfer learning tasks show that HASTE-modified metrics are consistently better or on par with the state-of-the-art transferability metrics. Our code is available here.

## 1 Introduction

Transfer learning (Pan & Yang, 2009; Torrey & Shavlik, 2010; Weiss et al., 2016) aims to improve the performance of models on target tasks by utilizing the knowledge from source tasks. With the increasing development of large-scale pre-trained models (Devlin et al., 2019; Chen et al., 2020a;b; Radford et al., 2021b), and the availability of multiple model choices (e.g model hubs of Pytorch, Tensorflow, Hugging Face) for transfer learning, it is critical to estimate their transferability without training on the target task and determine how effectively transfer learning algorithms will transfer knowledge from the source to the target task. To this end, transferability estimation metrics (Zamir et al., 2018b; Achille et al., 2019; Tran et al., 2019b; Pándy et al., 2022; Nguyen et al., 2020) have been recently proposed to quantify how easy it is to use the knowledge learned from these models with minimal to no additional training using the target dataset. Given multiple pre-trained source models and target datasets, estimating transferability is essential because it is non-trivial to determine which source model transfers best to a target dataset, and that training multiple models using all source-target combinations can be computationally expensive.

Recent years have seen a few different approaches (Zamir et al., 2018b; Achille et al., 2019; Tran et al., 2019b; Pándy et al., 2022; Nguyen et al., 2020) for estimating a given transfer learning task from a source model. However, existing such methods often require performing the transfer learning task for parameter optimization (Achille et al., 2019; Zamir et al., 2018b) or making strong assumptions on the source and target datasets (Tran et al., 2019b; Zamir et al., 2018b). In addition, they are limited to estimating transferability on specific source architectures (Pándy et al., 2022) or achieve lower performance when there are large domain differences between the source and target dataset (Nguyen et al., 2020). This has recently led to the questioning of the applicability of such metrics beyond specific settings (Agostinelli et al., 2022a).

Prior works in other contexts (Khan et al., 2018; Agarwal et al., 2022; Zhang et al., 2021b; Khan et al., 2018; Soviany et al., 2022; D'souza et al., 2021) show that machine learning (ML) models find some samples easier to learn while others are much harder. In this work, we observe and leverage a similar phenomenon in transfer learning tasks (Figure 1a), where images belonging to the harder subset of the target dataset achieve lower prediction accuracy than images from the easy subset. The

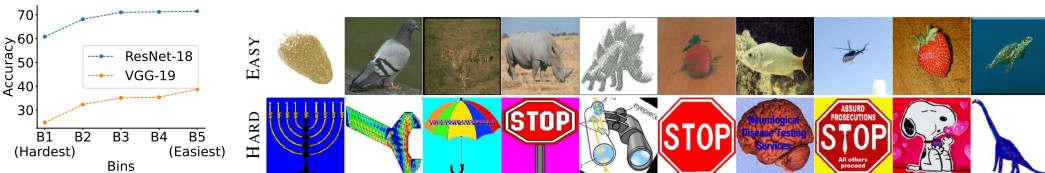

(a) Transfer Accuracy  (b) Top-10 images from Easy and Hard subsets

Figure 1: Analyzing the impact of hard subsets in transfer learning. **Column (a):** Results show the accuracy of different bins of a target dataset (Caltech101) based on their *hardness*. Across two source models (VGG-19 and ResNet-18) trained on the ImageNet dataset, we observe that the accuracy for images in the hardest subset (B1) is lower as compared to the easier subset (B5). **Column (b):** Top-10 images from hard and easy subsets show that harder subsets comprise images (cliparts) that are out-of-distribution when compared to the source dataset images. See Figures 5-9 for more qualitative images for different source-target pairs.

key principle is that easy samples do not contribute much when comparing the performance of a pre-trained model on multiple datasets or ranking the performance of different models on a given dataset. Additionally, in Figure 1b, we observe qualitatively that easy examples of the target dataset (Caltech101) comprise images that are in-distribution as compared to the source dataset (ImageNet), whereas images from the harder subset contain out-of-distribution clip art images that are not present in the source dataset and, hence, may be more challenging in the transfer learning process.

**Present work.** In this work, we incorporate the aforementioned observation and propose a novel framework, HASTE[1] (HArd Subset TransfErability), to estimate transferability by only using the hardest subset of the target dataset. More specifically, we introduce two complementary techniques – class-agnostic and class-specific – to identify harder subsets from the target dataset using the model's internal and output representations (Section 4.1). Further, we theoretically and empirically show that HASTE transferability metrics inherit the properties of its baseline metric and achieve tighter lower and upper bounds (Section 4.2).

We perform experiments across a range of transfer learning tasks like source architecture selection (Section 5.1), target dataset selection (Section 5.2), and ensemble model selection (Section 5.4), as well as on other tasks such as semantic segmentation (Section 5.3) and language models (Section 5.5). Our results show that HASTE scores better correlate with the actual transfer accuracy than their corresponding counterparts (Nguyen et al., 2020; Tran et al., 2019a; Pándy et al., 2022). Finally, we establish that our findings are agnostic to the choice of source architecture for identifying harder subsets, scale to transfer learning tasks for different data domains and that utilizing the hardest subsets can be highly beneficial for estimating transferability.

## 2   RELATED WORK

This work lies at the intersection of transfer learning and diverse metrics to estimate transferability from a source model to a target dataset. We discuss related works for each of these topics below.

**Transfer Learning (TL).** It can be organized into three broad categories: i) *Inductive Transfer* (Erhan et al., 2010; Yosinski et al., 2014), which leverages inductive bias, ii) *Transductive Transfer*, which is commonly known as Domain Adaptation (Wang & Deng, 2018; Wilson & Cook, 2020), and iii) *Task Transfer* (Zamir et al., 2018a; Pal & Balasubramanian, 2019), which transfers between different tasks instead of models. Amongst this, the most common form of a transfer learning task is fine-tuning a pre-trained source model for a given target dataset. For instance, recent works have demonstrated the use of large-scale pre-trained models such as CLIP (Radford et al., 2021a) and VirTex (Desai & Johnson, 2021) for learning representations for different source tasks.

**Transferability Metrics.** Despite the development of a plethora of source models, achieving an optimal transfer for a given target task is still a nascent research area as it is non-trivial to identify the source model or dataset for efficient TL. Transferability metrics are used as proxy scores to

---

[1]Code: https://anonymous.4open.science/r/haste/

estimate the transferability from a source to a target task. Prior works have proposed diverse metrics to estimate TL accuracy. For instance, NCE (Tran et al., 2019a) and LEEP (Nguyen et al., 2020) utilize the labels in the source and target task domains to estimate transferability. Further, metrics like H-Score (Bao et al., 2019), GBC (Pándy et al., 2022) and TransRate Huang et al. (2022) use the embeddings from the source model to estimate transferability. In contrast to the above metrics that focus on a single source model, Agostinelli et al. (2022b) explored metrics to estimate the transferability for an ensemble of models and introduced two metrics – MS-LEEP and E-LEEP – for identifying a subset of model ensembles from the pool of available source models.

## 3 PRELIMINARIES

**Notations.** Let a transfer learning task comprise of a pre-trained source model $f_\theta^s$ trained on a source dataset $\mathcal{D}_s = \{\mathcal{D}_s^{\text{train}}, \mathcal{D}_s^{\text{test}}\}$, and a target dataset $\mathcal{D}_t = \{\mathcal{D}_t^{\text{train}}, \mathcal{D}_t^{\text{test}}\}$ for transfer learning. We define a target model $f_\theta^{s \to t}$ which is initialized using the source model weights and are fine-tuned on the target dataset $\mathcal{D}_t^{\text{train}}$. The performance of the target model $f_\theta^{s \to t}$ is quantified using the target model accuracy $\mathcal{A}^{s \to t}$ when evaluated on the unseen target test dataset $\mathcal{D}_t^{\text{test}}$. Despite fine-tuning the source model, training target models is computationally expensive. Hence, we define a transferability metric $\mathcal{T}^{s \to t}$ which correlates with the target model accuracy $\mathcal{A}^{s \to t}$ and gives an efficient estimation of how the transfer learning will unfold for a given pair of source model and target dataset.

**Probability Estimations.** Let the source model $f_\theta^s$ output softmax scores over the source dataset label space $\mathcal{Z}$. Next, we construct a "source label distribution" of the target dataset over the source label space $\mathcal{Z}$ by passing them through $f_\theta^s$ and use it to build an empirical joint distribution over the source and target label spaces, i.e., $\hat{P}(y, z) = \frac{1}{n} \sum_{i:y_i=y} f_\theta^s(\mathbf{x}_i)_z$, where $f_\theta^s(\mathbf{x}_i)_z$ represents the softmax score of an instance $\mathbf{x}_i$ for class $z \in \mathcal{Z}$. Finally, the empirical marginal distribution and conditional distribution can be computed using $\hat{P}(z) = \sum_{y \in \mathcal{Y}} \hat{P}(y, z)$ and $\hat{P}(y|z) = \frac{\hat{P}(y,z)}{\hat{P}(z)}$, where $\mathcal{Y}$ denotes the target label space for dataset $\mathcal{D}_t$.

**Transferability Metric.** Following prior works, the performance of a transferability metric is evaluated by measuring the correlation between $\mathcal{T}^{s \to t}$ and $\mathcal{A}^{s \to t}$. Further, we focus on the *fine-tuning* style of transfer learning. Here, the final source classification layer of the source model $f_\theta^s$ is replaced with the target classification layer, and the whole model is trained on the target dataset task.

## 4 OUR METHOD: HASTE

Next, we describe, HASTE, a meta-transferability metric, which improves the transferability estimates by leveraging the harder subsets of the target data. We first discuss two complementary techniques (class-agnostic and class-specific; Sec. 4.1) to identify harder subsets and show that they can be applied to any of the existing transferability metrics. Next, we theoretically and empirically show that HASTE transferability metrics inherit the properties of their baseline metric (Sec. 4.2).

**Problem Statement (Transferability Metric).** *Given a source model $f_\theta^s$, source dataset $\mathcal{D}_s$, and target dataset $\mathcal{D}_t$, a transferability metric aims to output a score $\mathcal{T}^{s \to t}$ that correlates with the accuracy $\mathcal{A}^{s \to t}$ of the target model $f_\theta^{s \to t}$.*

### 4.1 CALCULATING HARDNESS

Here, we define the methods for identifying harder subsets in the target dataset, where one method uses the overall data distribution, and the other controls individual samples/classes to provide more representation of the dataset.

**Class-Agnostic Method.** The *Class-Agnostic* method uses the representation similarity between the samples in the source and target dataset to compute hardness scores. In particular, it employs embeddings from multiple layers of the source model and compares them for the source and target samples using cosine similarity, i.e.,

$$\psi(\mathbf{x}_i^s, \mathbf{x}_j^t) = \frac{1}{L} \sum_{l=1}^{L} \mathcal{E}_l(\mathbf{x}_i^s) \cdot \mathcal{E}_l(\mathbf{x}_j^t), \tag{1}$$

---

**Algorithm 1** HASTE

---

**Require:** Source model $f_\theta^s$, Source dataset $\mathcal{D}_s$, Target dataset $\mathcal{D}_t$, hard subset variant $h_v$, Transferability Metric $\mathcal{T}$
    $k \leftarrow$ Number of samples in hard subset
    **if** $h_v =$ 'ca' **then**
        Collect source dataset activations $\mathcal{E}_l(\mathbf{x}_i^s)$                              ▷ Class-Agnostic Case
        Collect target dataset activations $\mathcal{E}_l(\mathbf{x}_j^t)$
        Compute similarity matrix $\mathbf{S}_{ij} \leftarrow \psi(\mathbf{x}_i^s, \mathbf{x}_j^t)$                      ▷ As per Eqn. 1
        Compute Hardness $H(\mathbf{x}_j^t)$ for each target image using $\mathbf{S}_{ij}$
    **else if** $h_v =$ 'cs' **then**
        Collect target dataset activations $\mathcal{E}_l(\mathbf{x}_j^t)$                            ▷ Class-Specific Case
        **for** class $c \in \mathcal{D}_t$ **do**
            Compute $\mu_c, \Sigma_c$                                     ▷ As per Eqn. 3
        **end for**
        Compute Hardness $H(\mathbf{x}_j^t)$ for each target image using Mahalanobis Distance from $\mu_c$
    **end if**
    $\mathcal{D}_t^{\text{hard}} \leftarrow \{k \text{ hardest samples ordered by } H(\cdot)\}$
    **return** $\mathcal{T}(f_\theta^s, \mathcal{D}_t^{\text{hard}})$

---

where $\psi$ represents the similarity between a pair of source $\mathbf{x}_i^s$ and target $\mathbf{x}_j^t$ sample, $\mathcal{E}_l(\cdot)$ is the intermediate layer output from the $l$-th layer of $f_\theta^s$, and $L$ is the total number of layers in $f_\theta^s$. Next, we calculate an activation similarity matrix $\mathbf{S} \in \mathbb{R}^{M \times N}$, where $\mathbf{S}_{ij} = \psi(\mathbf{x}_i^s, \mathbf{x}_j^t)$, $M = |\mathcal{D}_s^{\text{train}}|$ and $N = |\mathcal{D}_t^{\text{train}}|$. Using the pairwise similarity matrix $\mathbf{S}$, we compute the *hardness* score for a target image, where samples *closer* to the source dataset obtain *lower* hardness scores, and vice-versa.

$$H(\mathbf{x}_j^t)_{CA} = 1 - \frac{1}{M}\sum_{i=1}^{M} \mathbf{S}_{ij}, \tag{2}$$

**Class-Specific Method.** In contrast to the class-agnostic strategy, which does not utilize label information of the target dataset, we introduce a *Class-Specific* technique to identify harder subsets by controlling the target classes (as they provide more representation of the dataset). Following Pándy et al. (2022), we model each target class $c$ as a normal distribution in the embedding space of $f_\theta^s$ and define the mean and covariance of the distribution as:

$$\mu_c = \frac{1}{N_c}\sum_{i:y_i^t=c} f_\theta^s(x_i^t); \quad \Sigma_c = \frac{1}{N_c}\sum_{i:y_i^t=c}(f_\theta^s(x_i) - \mu_c)(f_\theta^s(x_i) - \mu_c)^\top, \tag{3}$$

where $y_j^t = c$, and $N_c$ is the number of samples in class $c$. For each target sample, the hardness is defined as the Mahalanobis distance of the sample from the mean of the corresponding class distribution (Equation 3), i.e.,

$$H(\mathbf{x}_j^t)_{CS} = \sqrt{(f_\theta^s(x_i) - \mu_c)^\top \Sigma_c^{-1}(f_\theta^s(x_i) - \mu_c)} \tag{4}$$

Next, we use the above-mentioned techniques to identify hard subsets from the target dataset.

**HASTE.** To identify hard subsets, we sort the target dataset samples using either of the *hardness* scores, as defined in Equations 2,4. We denote the indices of the sorted samples using $\{q_1, q_2, \ldots q_N\}$. The hard subset is then defined as: $\mathcal{D}_t^{\text{hard}} = \{(\mathbf{x}_{q_1}^t, y_{q_1}^t), \ldots, (\mathbf{x}_{q_k}^t, y_{q_k}^t)\}; k \leq N$, where the hardness of each sample follows $H(\mathbf{x}_{q_1}^t) \geq H(\mathbf{x}_{q_2}^t) \geq \ldots H(\mathbf{x}_{q_N}^t)$. For using our HASTE modification with the existing metrics, we propose the use of only these identified harder subsets $\mathcal{D}_t^{\text{hard}}$ as an input to these metrics for estimating transferability, i.e.,

$$\text{HASTE} = \mathcal{T}(f_\theta^s, \mathcal{D}_t^{\text{hard}}), \tag{5}$$

where $\mathcal{T}(\cdot)$ denotes any existing transferability metric. See Algorithm 1 for details on getting harder subsets using Class Specific or Class Agnostic methods. Finally, we show the t-SNE (van der Maaten & Hinton, 2008) embeddings (Figure 2) of the entire target dataset and their harder subsets. We

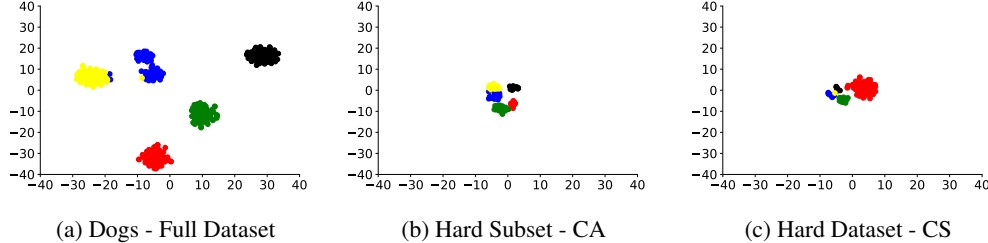

| (a) Dogs - Full Dataset | (b) Hard Subset - CA | (c) Hard Dataset - CS |

Figure 2: t-SNE embeddings of the entire target dataset and its hardest subset using a ResNet-50 source model trained on ImageNet. We show the embeddings from five random classes from Stanford Dogs as the target dataset, using both the class-agnostic (b) and class-specific (c) methods. We observe that the embeddings from the harder subset are more entangled than the entire dataset.

observe that the embeddings from the entire dataset (Figure 2a) are well segregated, but embeddings of samples in the harder subsets are highly entangled (Figure 2b-2c). These findings align with our findings in Figure 1, where images from harder subsets achieve lower transfer accuracies, i.e., the source models struggle to find the decision boundaries between these harder samples.

## 4.2 THEORETICAL PROPERTIES

Here, we show that HASTE-modified metrics inherit the theoretical properties of their baseline metric. Note that showing theoretical bounds for all transferability metrics is outside the scope of this work. Hence, we take one representative metric (LEEP) and show that HASTE-LEEP retains its theoretical properties.

**LEEP.** Let source model $f_\theta^s$ predict the target label $y$ by directly drawing from the label distribution $p(y|\mathbf{x}; f_\theta^s, \mathcal{D}_t^{\text{train}}) = \sum_{z \in \mathcal{Z}} \hat{P}(y|z) f_\theta^s(\mathbf{x})_z$. The LEEP score is then defined as average log-likelihood:

$$\text{LEEP} = \mathcal{T}(f_\theta^s, \mathcal{D}_t^{\text{train}}) = \frac{1}{n} \sum_{i=1}^n \log \big( \sum_{z \in \mathcal{Z}} \hat{P}(y|z) f_\theta^s(\mathbf{x})_z \big) \tag{6}$$

**Average log-likelihood.** We fix the source model weights $\theta$ and re-train the classification model using maximum likelihood and the target dataset $\mathcal{D}_t^{\text{train}}$ to obtain a new classifier $f_\theta^*$, i.e.,

$$f_\theta^* = \arg\max_{k \in \mathcal{K}} l(\theta, k), \tag{7}$$

where $l(\theta, k)$ is the average likelihood for the weights $\theta$ and $k$ on the target dataset $\mathcal{D}_t^{\text{train}}$, and $k$ is selected from a space of classifiers $\mathcal{K}$.

**Lemma 1.** HASTE-LEEP *is a lower bound of the optimal average log-likelihood for the hard subset.*

$$\mathcal{T}(f_\theta^s, \mathcal{D}_t^{hard}) \leq l(w, k^*)^{hard} \leq l(w, k^*) \tag{8}$$

*Proof.* This proof is true by definition as $\mathcal{D}_t^{\text{hard}} \subset \mathcal{D}_t^{\text{train}}$ represents the hard subset of the target dataset. Note that $l(w, k^*)$ is the maximal average log-likelihood over $k \in K$, and $\mathcal{T}(f_\theta^s, \mathcal{D}_t^{\text{train}})$ is the average log-likelihood in $K$. From Nguyen et al. (2020) we know $\mathcal{T}(f_\theta^s, \mathcal{D}_t^{\text{train}}) \leq l(w, k^*)$ and by definition of $\mathcal{D}_t^{\text{hard}}$, $\mathcal{T}(f_\theta^s, \mathcal{D}_t^{\text{hard}}) \leq l(w, k^*)$. In addition, the model struggles to learn the samples in the hard subset, and hence $l(w, k^*)^{\text{hard}} \leq l(w, k^*)$ □

**Lemma 2.** HASTE-LEEP *is an upper bound of the NCE measure plus the average log-likelihood of the source label distribution, computed over the hard subset, i.e.,*

$$\mathcal{T}(f_\theta^s, \mathcal{D}_t^{hard}) \geq \textit{H-NCE}(Y \mid Z) + \frac{1}{|\mathcal{D}_t^{hard}|} \Sigma_{i=1}^{|\mathcal{D}_t^{hard}|} \log f_\theta^s(\mathbf{x}_i)_{z_i}, \tag{9}$$

*Proof Sketch.* This proof extends from the property of LEEP. See Appendix A.1 for detailed proof. □

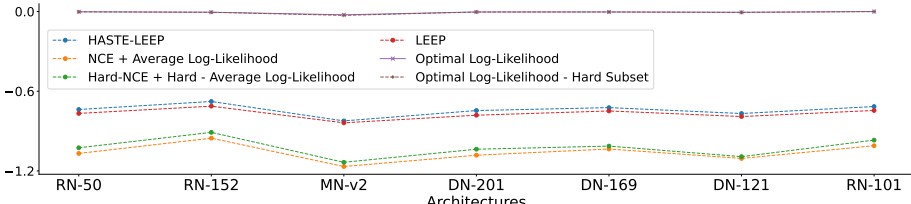

Figure 3: Empirical results on the StanfordDogs target dataset show no violations of our theoretical bounds. Empirically calculated HASTE-LEEP (in blue) and our theoretical upper (in purple) and lower (in green) bounds from Equations 8-9 across seven source model architectures trained on the ImageNet dataset, where RN = ResNet, MN = MobileNet, DN = DenseNet.

**Empirical Analysis.** We analytically evaluated the upper and lower bounds for HASTE-LEEP by computing the RHS of Equations 8-9. In Figure 3, our results show HASTE-LEEP and its corresponding theoretical upper and lower bounds, confirming that, across seven source model architectures, none of our theoretical bounds are violated. In addition, we empirically demonstrate that our bounds are tighter than LEEP.

## 5 EXPERIMENTS

Next, we present experimental results to show the effectiveness of HASTE modified transferability metrics for different transfer learning tasks, including source architecture selection (Sec. 5.1), target dataset selection (Sec. 5.2), semantic segmentation task (Sec. 5.3), ensemble model selection (Sec. 5.4), and language domain transferability (Sec. 5.5).

**Evaluation metrics and Baselines.** We use the Pearson Correlation Coefficient (PCC) for correlation between $\mathcal{T}^{s \to t}$ and $\mathcal{A}^{s \to t}$ (see Tables 10-11 for results of other correlation coefficients). For baselines, we use LEEP, NCE, and GBC for single model transferability tasks, and MS-LEEP and E-LEEP for ensemble model selection. See Appendix A.2 for more details.

### 5.1 SOURCE ARCHITECTURE SELECTION

**Experimental setup.** As detailed inPándy et al. (2022), the target dataset is fixed and the $\mathcal{T}^{s \to t}$ are computed over multiple source architectures. The correlation scores are computed between $\mathcal{T}^{s \to t}$ and the transfer accuracies $\mathcal{A}^{s \to t}$. We consider seven target datasets for our source architecture experiments: i) Caltech101 (Fei-Fei et al., 2004), ii) CUB200 (Welinder et al., 2010), iii) Oxford-IIIT Pets (Parkhi et al., 2012), iv) Flowers102 (Nilsback & Zisserman, 2008), v) Stanford Dogs (Khosla et al., 2011), vi) Imagenette (Howard), and vii) PACS-Sketch (Li et al., 2017).

**Model architectures and Training.** We consider seven source architectures pre-trained on ImageNet (Russakovsky et al., 2015) dataset, including ResNet-50, ResNet-101, ResNet-152 (He et al., 2016), DenseNet-121, DenseNet-169, DenseNet-201 (Huang et al., 2017), and MobileNetV2 (Sandler et al., 2018). All models were set using the publicly available pre-trained weights from the Torchvision library (Marcel & Rodriguez, 2010). For each source architecture, we utilize the ResNet-50 model to calculate the hardness ranking, and the hard subset used to compute HASTE scores. Following Pándy et al. (2022), we calculate the target accuracy $\mathcal{A}^{s \to t}$ by *fine-tuning* the source model on each target dataset. We fine-tune the source model for 100 epochs using an SGD optimizer with a momentum of 0.9, a learning rate of $10^{-4}$, and a batch size of 64.

**Results.** On average, across seven target datasets, results show an improvement in correlation scores of +129.74% for LEEP, +29.38% for NCE, and -0.07% GBC, using HASTE-modified metrics (Table 1). Interestingly, for most target datasets, both CA and CS variants of the HASTE metrics outperform the baseline scores.

### 5.2 TARGET DATASET SELECTION

**Experimental setup.** Here, the source model is fixed and the transferability metric is computed over multiple target datasets (Nguyen et al., 2020). We construct 50 target datasets by randomly selecting

Table 1: Results on source architecture selection task. Shown are correlation scores (higher the better) computed across all source architectures trained on ImageNet. Results where HASTE modified metrics perform better than their counterparts are in **bold**.

| Target ($\mathcal{D}_T$) | LEEP | HASTE-LEEP | | NCE | HASTE-NCE | | GBC | HASTE-GBC | |
| --- | --- | --- | --- | --- | --- | --- | --- | --- | --- |
| | | CA | CS | | CA | CS | | CA | CS |
| CUB200 | 0.534 | 0.405 | **0.667** | 0.330 | 0.040 | **0.500** | 0.790 | **0.811** | 0.785 |
| StanfordDogs | 0.926 | **0.943** | **0.931** | 0.930 | 0.924 | **0.955** | 0.784 | **0.944** | **0.834** |
| Flowers102 | 0.504 | **0.508** | **0.723** | 0.382 | **0.390** | **0.388** | -0.012 | -0.013 | -0.02 |
| Oxford-IIIT | 0.921 | **0.952** | **0.927** | 0.846 | **0.851** | **0.916** | 0.668 | **0.867** | **0.745** |
| Caltech101 | 0.416 | **0.439** | **0.458** | 0.204 | **0.461** | **0.504** | 0.810 | 0.793 | **0.821** |
| Imagenette | 0.950 | **0.950** | **0.962** | 0.927 | **0.940** | 0.889 | 0.709 | **0.723** | **0.711** |
| PACS-Sketch | -0.029 | **0.196** | **0.253** | -0.129 | **0.160** | -0.208 | 0.612 | **0.637** | 0.601 |

Table 2: Results on target task selection using the fine-tuning method for Caltech101 source models. Shown are correlation scores (higher the better) computed across all target datasets. Results, where HASTE modified metrics perform better than their counterparts, are in **bold**. See Table 6 for results on CUB200 source models.

| Target ($\mathcal{D}_t$) | LEEP | HASTE-LEEP | | NCE | HASTE-NCE | | GBC | HASTE-GBC | |
| --- | --- | --- | --- | --- | --- | --- | --- | --- | --- |
| | | CA | CS | | CA | CS | | CA | CS |
| CUB200 | 0.948 | **0.950** | 0.948 | 0.944 | **0.948** | 0.944 | 0.916 | **0.917** | 0.916 |
| Flowers102 | 0.769 | **0.820** | 0.761 | 0.762 | **0.823** | 0.758 | 0.743 | 0.742 | 0.727 |
| StanfordDogs | 0.884 | **0.901** | 0.884 | 0.885 | **0.899** | **0.886** | 0.873 | **0.876** | 0.856 |
| Oxford-IIIT | 0.899 | **0.907** | **0.905** | 0.899 | **0.905** | **0.908** | 0.845 | **0.854** | **0.858** |
| PACS-Sketch | 0.940 | **0.943** | **0.944** | 0.939 | **0.940** | **0.941** | 0.692 | **0.852** | **0.894** |

a subset of classes from the original target dataset. The PCC is computed between $\mathcal{T}^{s \to t}$ and $\mathcal{A}^{s \to t}$ across all 50 target tasks, where each target subset contains 40% to 100% of the total classes, and for each class, all train and test images are included in the subset. We consider six target datasets including Caltech101, CUB200, Oxford-IIIT Pets, Flowers102, Stanford Dogs, and PACS-Sketch.

**Model architectures and Training.** We consider two source models: ResNet-18 pre-trained on CUB200 and ResNet-34 pre-trained on Caltech101. We train the transferred models for 100 epochs using SGD with a momentum of 0.9, a learning rate of $10^{-3}$, and a batch size of 64.

**Results.** Across two source datasets and four target datasets, HASTE-LEEP achieves the highest correlation for the target selection task, and outperform their respective baseline methods (Table 2). In particular, we observe an improvement in correlation scores of +0.99% for LEEP, +1.15% for NCE, and +5.11% for GBC.

### 5.3 SEMANTIC SEGMENTATION

**Experimental setup.** We follow the fixed target setting described in Pándy et al. (2022) and report the correlation between meanIoU and $\mathcal{T}^{s \to t}$ for each target dataset. We consider a Fully Connected Network (FCN) Long et al. (2014) with a ResNet-50 backbone pre-trained on a subset of COCO2017 (Lin et al., 2014). We consider CityScapes (Cordts et al., 2016), CamVid (Brostow et al., 2009), BDD100k (Yu et al., 2018), IDD (Varma et al., 2019), PascalVOC (Li et al., 2020) and SUIM (Islam et al., 2020) datasets. Among them, we consider the target datasets CityScapes (Cordts et al., 2016), CamVid (Brostow et al., 2009), BDD100k (Yu et al., 2018) and SUIM (Islam et al., 2020). Note that we use the CA variant of HASTE for semantic segmentation as segmentation does not have class labels.

**Model architectures and Training.** We train an FCN Resnet50 Long et al. (2014) model for each source training dataset (except the target) and individually fine-tune them on $\mathcal{D}_t^{\text{train}}$. We train the individual models for 100 epochs using SGD with a momentum of 0.9, weight decay of $10^{-4}$, a batch size of 16, a learning rate of $10^{-3}$, and reduce it on plateau by a factor of 0.5. Each model is fine-tuned on the target dataset independently, using SGD with a momentum of 0.9, a learning rate of $10^{-3}$, and a batch size of 16.

**Results.** On average across four target datasets, results show that HASTE-modified metrics outperform their baseline methods (Table 3). In particular, we observe an improvement in correlation scores of +182.23% for LEEP, +33.34% for NCE, and +149.30% for GBC.

Table 3: Results on the semantic segmentation source architecture selection task. Shown are correlation scores (higher the better) computed across all source architectures. Results where HASTE modified metrics perform better than their counterparts are in **bold**.

| Target ($\mathcal{D}_t$) | LEEP | HASTE-LEEP | NCE | HASTE-NCE | GBC | HASTE-GBC |
|---|---|---|---|---|---|---|
| BDD100k | 0.147 | **0.197** | 0.731 | **0.743** | 0.645 | **0.660** |
| CamVid | 0.063 | **0.374** | 0.573 | **0.583** | 0.334 | **0.796** |
| SUIM | 0.823 | **0.980** | 0.204 | **0.461** | -0.218 | **0.784** |
| CityScapes | 0.045 | **0.127** | 0.524 | **0.545** | 0.952 | 0.923 |

Table 4: Results on the ensemble model selection task. Shown are correlation scores (higher the better) computed across all ensemble candidates. Results where HASTE modified metrics perform better than their counterparts are in **bold**. See Appendix Table 7 for results using $K=3$.

| Target ($\mathcal{D}_t$) | MS-LEEP | HASTE-MS-LEEP CA | HASTE-MS-LEEP CS | E-LEEP | HASTE-E-LEEP CA | HASTE-E-LEEP CS |
|---|---|---|---|---|---|---|
| Flowers102 | 0.230 | **0.368** | **0.251** | 0.271 | **0.314** | 0.244 |
| Stanford Dogs | 0.400 | 0.378 | 0.400 | 0.503 | **0.522** | **0.506** |
| CUB200 | 0.334 | **0.411** | 0.324 | 0.402 | **0.403** | **0.434** |
| OxfordPets | 0.112 | **0.148** | **0.133** | 0.276 | **0.338** | **0.281** |
| Caltech101 | 0.462 | **0.502** | **0.467** | 0.520 | 0.513 | 0.518 |

## 5.4 ENSEMBLE MODEL SELECTION

**Experimental setup.** Given a pool with $P$ number of source models, this task aims to select the subset of models whose ensemble yields the best performance on a fixed target dataset (Agostinelli et al., 2022b). Since evaluating every ensemble combination of the $P$ source models is very expensive, the ensemble size $K$ (i.e., number of models per ensemble) is fixed, which yields $\binom{P}{K}$ candidate ensembles. The PCC is then computed between the $\mathcal{T}^{s \to t}$ and $\mathcal{A}^{s \to t}$ across all candidate ensemble. We use $K=4$ and $P=11$ in our experiments and consider the target datasets from Section 5.2.

**Model architectures and Training.** We include source models pre-trained on the above datasets as well as ImageNet. Each ensemble of model architectures consists of one or more models from the pool of ResNet-101, VGG-19 (Simonyan & Zisserman, 2015), and DenseNet-201 with each model pre-trained on the mentioned datasets. For a given candidate ensemble, each member model is fine-tuned individually on a fixed target train dataset $\mathcal{D}_t^{\text{train}}$, and, finally, the ensemble prediction is calculated as the mean of all individual predictions. Each model is fine-tuned on the target dataset independently, using SGD with a momentum of 0.9, a learning rate of $10^{-4}$, and a batch size of 64.

**Results.** Our empirical analysis in Table 4 shows that, on average across five target datasets, HASTE-modified metrics outperform their baseline methods. In particular, we observe an improvement in correlation scores of +14.43% for MS-LEEP and +4.10% for E-LEEP.

## 5.5 ADDITIONAL RESULTS ON LANGUAGE MODELS

**Experimental setup.** We now evaluate the performance of HASTE for the sentiment classification transfer learning tasks and show results in the target dataset selection setting. We consider three target datasets, including TweetEval (Barbieri et al., 2018), IMDB Movie Reviews (Maas et al., 2011), and CARER (Saravia et al., 2018) for our language experiments.

**Model architecture and Training.** We include source models trained using a classification head on a pre-trained BERT (Devlin et al., 2019) model on CARER (Saravia et al., 2018) and AG-News (Zhang et al., 2015) datasets. We fine-tune the entire source model, including the BERT layers for 3 epochs using the Adam optimizer, with a learning rate of $5 \times 10^{-5}$, and a batch size of 8.

**Results.** Table 5 show that HASTE-modified transferability metrics outperform their baseline counterparts. On average across four source-target pairs and two techniques, we observe an improvement of +38.13% for LEEP, +33.40% for NCE, and +57.24% for GBC using HASTE metrics.

## 5.6 ABLATION STUDIES

We conduct ablations on two key components of HASTE modified transferability metrics: i) size of the hardest subset and ii) correlation estimates using different buckets. We also study the impact

Table 5: Results on target task selection for sentiment classification. Shown are correlation scores (higher the better) computed across all target candidates. Results where HASTE modified metrics perform better than their counterparts are in **bold**.

| Source-Target Pair | LEEP | HASTE-LEEP | | NCE | HASTE-NCE | | GBC | HASTE-GBC | |
|---|---|---|---|---|---|---|---|---|---|
| | | CA | CS | | CA | CS | | CA | CS |
| Emotion - IMDB | -0.172 | **0.115** | **0.06** | -0.192 | **0.073** | **0.050** | -0.097 | **0.141** | **0.109** |
| Emotion - TweetEval | 0.884 | **0.892** | **0.885** | 0.884 | **0.892** | **0.885** | 0.828 | **0.834** | 0.824 |
| AGNews - Emotion | 0.939 | **0.943** | **0.944** | 0.940 | **0.947** | **0.944** | 0.808 | 0.808 | 0.808 |
| AGNews - TweetEval | 0.776 | **0.779** | **0.784** | 0.884 | **0.892** | **0.885** | 0.549 | 0.549 | 0.549 |

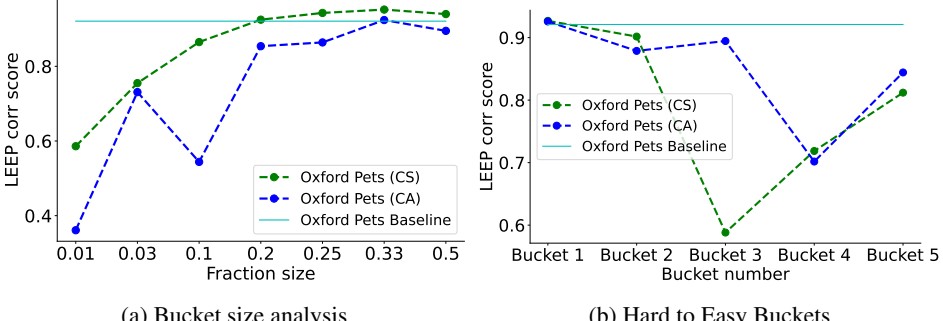

(a) Bucket size analysis          (b) Hard to Easy Buckets

Figure 4: Ablation results for HASTE. **Left:** It demonstrates the LEEP results (y-axis) on varying the size of the hard subset as a fraction of the complete dataset (x-axis) and shows that 25%-33% gives the best LEEP score. **Right:** It shows the LEEP results (y-axis) for hard-to-easy buckets (x-axis) and shows that the transferability scores are the highest for Bucket 1 (hardest).

of different architectures for computing hardness on the performance of the HASTE metrics (see Appendix A.4).

**Bucket Size Analysis.** Here, we follow the experimental setup from the source architecture selection (Section 5.1) and choose different sizes of the hardest bucket (or simply hardest subset). We vary the bucket size $b$ by using different fractions of the entire dataset and report the results for $b=\{0.01, 0.03, 0.1, 0.2, 0.25, 0.33, 0.5\}$. We find that, on average, the correlation performance increases as we increase the bucket size (Figure 4a). Further, bucket sizes in the range $b=[0.2, 0.4]$ generally give the best transferability estimates. This is intuitive because the influence of true hard samples might decrease in the light of easier samples for large bucket sizes, thus, going against the notion of HASTE, while for very small bucket sizes, the number of samples might not be enough for metrics like LEEP to show their effectiveness.

**Transferability along Hard to Easy Buckets.** A key question in HASTE is to understand the effect of different subsets (depending on their easiness or hardness) on estimating transferability. Here, we follow the experimental setup from source architecture selection (Section 5.1), calculate HASTE-LEEP using different buckets, and compare it with the baseline LEEP score using the entire dataset. Results show that transferability estimates are the best for harder subsets and gradually degrade while moving towards easier subsets (Figure 4b), confirming the core hypothesis of HASTE.

## 6 CONCLUSION

We propose and address the problem of estimating transferability from a source to the target domain using examples from the harder subset of the target dataset. To this end, we introduce HASTE (HArd Subset TransfErability) which leverages class-agnostic and class-specific strategies to identify harder subsets from a target dataset and can be used with any existing transferability metric. We show that HASTE-modified transferability metrics outperform their counterparts across different transfer learning tasks, data modalities, models, and datasets. In contrast to the findings in Agostinelli et al. (2022b), i.e., one metric doesn't work for all transfer learning tasks, we show that HASTE metrics achieve favorable results across diverse transfer learning settings (Sec. 5). Hence, we anticipate that using HASTE could open new frontiers in estimating transferability and pave way for several exciting future directions, like developing new techniques to identify harder subsets and extending HASTE analysis to other transfer learning tasks.

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

# A APPENDIX

## A.1 PROOF FOR LEMMA 2

**Lemma 2.** HASTE-LEEP *is an upper bound of the NCE measure plus the average log-likelihood of the source label distribution, computed over the hard subset, i.e.,*

$$\mathcal{T}(f_\theta^s, \mathcal{D}_t^{hard}) \geq \text{H-NCE}(Y \mid Z) + \frac{1}{|\mathcal{D}_t^{hard}|} \Sigma_{i=1}^{|\mathcal{D}_t^{hard}|} \log f_\theta^s(\mathbf{x}_i)_{z_i}, \tag{10}$$

*Proof.* Let $z_i$ be the dummy labels obtained when computing NCE and $y_i$ be the true labels.

$$\mathcal{T}(f_\theta^s, \mathcal{D}_t^{hard}) = \frac{1}{|\mathcal{D}_t^{hard}|} \sum_{i=1}^{|\mathcal{D}_t^{hard}|} \log \left( \sum_{z \in \mathcal{Z}} \hat{P}(y_i|z) f_\theta^s(\mathbf{x}_i)_z \right)$$

$$\geq \frac{1}{|\mathcal{D}_t^{hard}|} \Sigma_{i=1}^{|\mathcal{D}_t^{hard}|} \log \left( \hat{P}(y_i|z_i) f_\theta^s(\mathbf{x}_i)_{z_i} \right) \qquad \text{(Monotonicity of Log)}$$

$$= \frac{1}{|\mathcal{D}_t^{hard}|} \Sigma_{i=1}^{|\mathcal{D}_t^{hard}|} \log \hat{P}(y_i|z_i) + \frac{1}{|\mathcal{D}_t^{hard}|} \Sigma_{i=1}^{|\mathcal{D}_t^{hard}|} \log f_\theta^s(\mathbf{x}_i)_{z_i}$$

$$= \text{H-NCE}(Y \mid Z) + \frac{1}{|\mathcal{D}_t^{hard}|} \Sigma_{i=1}^{|\mathcal{D}_t^{hard}|} \log f_\theta^s(\mathbf{x}_i)_{z_i}.$$

$\square$

## A.2 EXPERIMENTAL SETUP

**Implementation details.** All experiments were run using the PyTorch library (Paszke et al., 2019) with Nvidia A-100/V-100 GPUs.

**Model Architectures.** We use a variety of model architectures (VGG, ResNet, DenseNet), trained on different source datasets across our experiments. For each model architecture, we utilize embeddings from the final layer for the class-specific method (Eqn. 3). For the class-agnostic method, we utilize embeddings from intermediate layers for the similarity computation (Eqn. 1). Particularly, we utilize the embeddings from the final layer of each block of convolutions (Ex: Output of each residual block in ResNets). We only include 3/4 layers for any architecture and do not consider embeddings from the first block.

**Similarity Computation for Large Source Datasets.** For the experiments with models pre-trained on ImageNet as the source, when using the class-agnostic method, it is infeasible to use the entire ImageNet dataset for the similarity comparison to generate the similarity matrix (using Eqn. 1). Instead, we use a random 10% subset of the ImageNet dataset, uniformly sampled from each class, as the source dataset to compute the similarity matrix. We do not observe any performance drop due to this sub-sampling, and this can be extended to other datasets as well, for further computational speedup. Additionally, we do not re-do the similarity computation for each source architecture. Instead, we only compute the similarity matrix using the ResNet-50 model and use the hard subset obtained from this to compute HASTE modified transferability metrics for all model architectures pre-trained on ImageNet. We repeat this in the model ensemble setting, utilizing a single similarity matrix for all model architectures trained on the same source dataset.

**GBC Implementation.** In all experiments, for computing GBC and HASTE-GBC, we use a spherical covariance matrix, as we found this to yield better results, even for the base GBC score.

**Size of Hard Subset.** The size of the hard subset in any experiment is a hyperparameter that can be tuned. Due to variations in dataset sizes, the exact value differs significantly. Instead of fixing a size, we set the size of the hard subset to be k% of the size of the target dataset. In general, we found a value of 10%-25% to work well.

**Subset for Ensemble Selection.** The hard subset for the Class Agnostic method is dependent on the source dataset. As a result, the experiment setting described in Section 5.4 has different hard subsets depending on the source dataset and model for a single target dataset. Since MS-LEEP is simply the addition of LEEP scores, the calculation of HASTE-MS-LEEP is trivial. But in the case of

HASTE-E-LEEP, we need a single common subset as it involves the calculation of adding empirical probabilities followed by log and mean. To this end, we take the union of the hard subsets obtained from different sources and then proceed with further calculations. We report the results in Table 4 following the same.

**Models used in Ensemble Selection.** We use the following pool of source models for ensemble selection experiment described in Section 5.4: i) DenseNet-201, ResNet-101, MobileNetv2 trained on on Imagenet, ii) DenseNet-201, ResNet-18, VGG-19 trained on Stanford Dogs, iii) DenseNet-201, ResNet-101 trained on Ocford IIIT Pets, iv) DenseNet-201, ResNet-101 trained on Flowers102, v) ResNet-18, VGG-19 trained on CUB200, and vi) ResNet-34 trained on Caltech101.

## A.3 ADDITIONAL RESULTS

**Computational cost.** To provide an indicative reference, we compare the run times of several transferability metrics, as well as our HASTE modification using a ResNet-18 trained on CUB200 as the source model (on a single GPU), and Oxford-IIIT Pets as the target dataset. Including the target dataset feature-extraction stage (shared by all metrics, as well as the class-agnostic method of HASTE), LEEP runs in 7.24s, NCE runs in 7.26s, and GBC runs in 8.33s. The class-agnostic variant on the HASTE modification takes an additional 24.98s, and the class-specific variant takes an additional 0.56s.

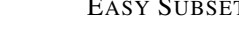
EASY SUBSET

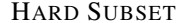
HARD SUBSET

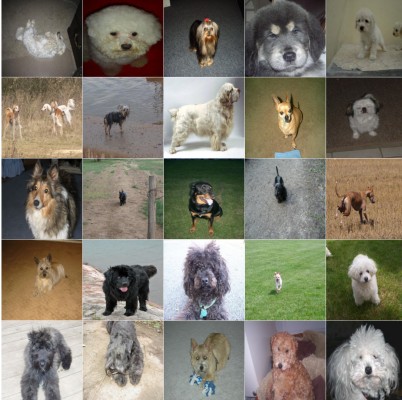
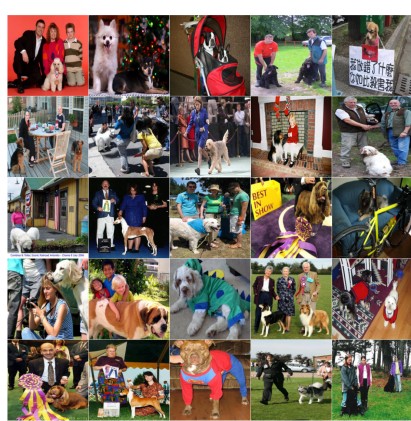

Figure 5: The $5 \times 5$ grid shows the top-25 images from the easy (left) and hard (right) subset of the target dataset using the class-agnostic technique for the ImageNet-StanfordDogs source-target pair. Images with *higher* hardness scores tend to feature cluttered images with atypical vantage points, whereas images with *lower* hardness scores mostly comprise dogs in an uncluttered background.

Table 6: Results on target task selection using the fine-tuning method for CUB200 source models. Shown are correlation scores (higher the better) computed across all target datasets. Results where HASTE modified metrics outperform their baselines are **bolded**.

| Target ($\mathcal{D}_t$) | LEEP | H-LEEP | | NCE | H-NCE | | GBC | H-GBC | |
|---|---|---|---|---|---|---|---|---|---|
| | | CA | CS | | CA | CS | | CA | CS |
| Caltech101 | -0.035 | **0.709** | **0.098** | 0.081 | **0.742** | **0.249** | 0.507 | **0.562** | **0.516** |
| Flowers102 | 0.612 | **0.617** | **0.613** | 0.593 | 0.579 | **0.618** | 0.535 | 0.526 | **0.568** |
| StanfordDogs | 0.929 | **0.936** | 0.929 | 0.928 | 0.927 | **0.929** | 0.909 | **0.914** | **0.913** |
| Oxford-IIIT | 0.863 | **0.871** | 0.860 | 0.812 | **0.826** | **0.814** | 0.859 | **0.860** | **0.872** |
| PACS-Sketch | 0.947 | **0.965** | **0.960** | 0.949 | **0.958** | **0.950** | 0.819 | **0.909** | **0.883** |

EASY SUBSET          HARD SUBSET

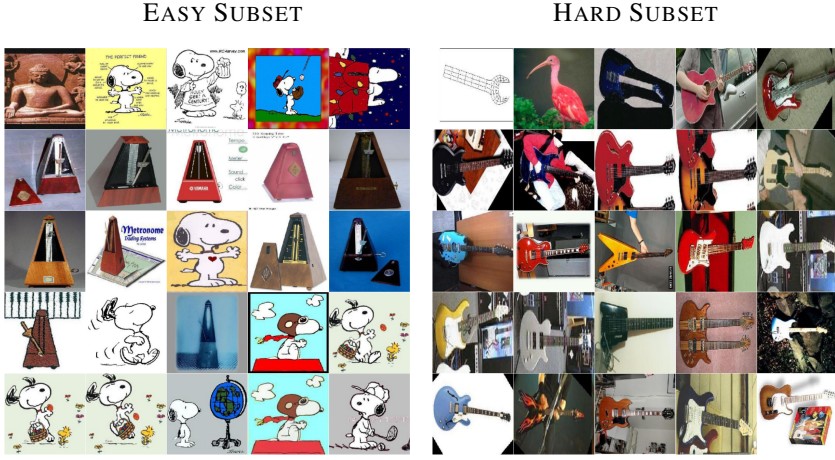

Figure 6: The $5 \times 5$ grid shows the top-25 images from the easy and hard subset of the target dataset using the class-agnostic technique for the ImageNet-OxfordIIIT Pets source-target pair. Images with *higher* hardness scores tend to feature cluttered images with atypical vantage points, whereas images with *lower* hardness scores mostly comprise dogs and cats in an uncluttered background.

EASY SUBSET          HARD SUBSET

Figure 7: The $5 \times 5$ grid shows the top-25 images from the easy (left) and hard (right) subset of the target dataset using the class-specific technique for the ImageNet-Caltech101 source-target pair. Images with *higher* hardness scores tend to feature classes that are typically harder to classify (since they might have very less distinguishing features), whereas images with *lower* hardness scores mostly comprise classes that are easily distinguishable.

Table 7: Results on the ensemble model selection task for $K = 3$. Shown are correlation scores (higher the better) computed across all ensemble candidates. Results where HASTE modified metrics outperform their baselines are **bolded**.

| Target ($\mathcal{D}_t$) | MS-LEEP | H-MS-LEEP | E-LEEP | H-E-LEEP |
|---|---|---|---|---|
| Flowers102 | -0.288 | -0.376 | -0.323 | **-0.319** |
| Stanford Dogs | 0.390 | 0.264 | 0.477 | **0.494** |
| CUB200 | 0.345 | **0.391** | 0.405 | 0.405 |
| Oxford-IIIT | 0.115 | **0.189** | 0.253 | **0.343** |
| Caltech101 | 0.430 | **0.479** | 0.480 | 0.478 |

EASY SUBSET                              HARD SUBSET

EASY SUBSET                              HARD SUBSET

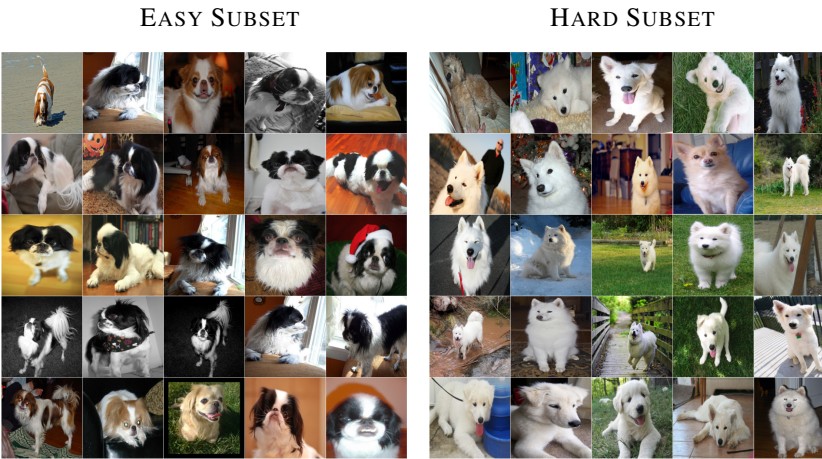

Figure 8: The $5 \times 5$ grid shows the top-25 images from the easy (left) and hard (right) subset of the target dataset using the class-specific technique for the ImageNet-StanfordDogs source-target pair. Images with *higher* hardness scores tend to feature classes that are typically harder to classify (since they might have very less distinguishing features), whereas images with *lower* hardness scores mostly comprise classes that are easily distinguishable.

Figure 9: The $5 \times 5$ grid shows the top-25 images from the easy (left) and hard (right) subset of the target dataset using the class-specific technique for the ImageNet-OxfordIIIT Pets source-target pair. Images with *higher* hardness scores tend to feature classes that are typically harder to classify (since they might have very less distinguishing features), whereas images with *lower* hardness scores mostly comprise classes that are easily distinguishable.

## A.4    ABLATION ON HARDNESS SOURCE ARCHITECTURE

HASTE aims to achieve better transferability estimates irrespective of the source of the hardness scores, i.e., the source architecture we use to calculate harder subsets in Class Agnostic way or Class Specific way. We follow the experimental setup from the target task selection (Section 5.2) experiments. We calculate HASTE-LEEP scores on harder subsets identified using i) ResNet18 and VGG19 trained on CUB200, and ii) ResNet50 trained on ImageNet. Results show that HASTE-LEEP outperforms LEEP (baseline calculated using the entire dataset) across all three architectures (Table 8).

Table 8: Results on target task selection task different source model architectures. Shown are correlation scores (higher the better) computed across the target dataset using HASTE-LEEP. Irrespective of the architecture used for finding the hard subset, HASTE-LEEP outperforms the baseline LEEP score.

| Hardness Source Model | Caltech101 | | Oxford-IIIT Pets | |
|---|---|---|---|---|
| | CA | CS | CA | CS |
| ResNet18 | 0.360 | 0.014 | 0.896 | 0.901 |
| VGG19 | 0.200 | 0.267 | 0.881 | 0.894 |
| ResNet50 | 0.630 | 0.196 | 0.896 | 0.898 |
| Baseline (LEEP Score) | -0.03 | | 0.863 | |

## A.5 EASIER SAMPLES ADDED WITH STOCHASTICITY

A natural question which may arise when using HASTE is whether to completely neglect the easier samples. To this end, we conduct an experiment where we add easier samples stochastically to our hard subsets and then compute the respective metric correlation score. We follow the same experiment setting as Section 1. We obtain results by iterating the addition of these easier samples 10 times followed by taking the mean. We observe that addition of these easier samples do not particularly enhance the results. In fact, the results come out to be worse than when using only hard subsets. Results for the same are shown in Table 9

Table 9: Results on source architecture selection task for subset obtained by stochastically adding easier samples to the hard subset. Shown are correlation scores (higher the better) computed across all source architectures trained on ImageNet. Results where HASTE modified metrics outperform their baselines are in **bold**.

| Target ($\mathcal{D}_t$) | LEEP | H-LEEP | | Stochasticity % | | | | | |
|---|---|---|---|---|---|---|---|---|---|
| | | CA | CS | 1% | 2% | 3% | 4% | 5% | 10% |
| Caltech101 | 0.416 | **0.439** | **0.475** | 0.474 | 0.468 | 0.472 | 0.472 | 0.472 | 0.458 |
| Flowers102 | 0.534 | 0.405 | **0.626** | 0.616 | 0.596 | 0.579 | 0.576 | 0.575 | 0.539 |
| CUB200 | 0.504 | **0.508** | **0.723** | 0.719 | 0.714 | 0.714 | 0.723 | 0.728 | 0.679 |

## A.6 RESULTS WITH ALTERNATE CORRELATION METRIC

HASTE approach is agnostic to the choice of correlation coefficient. We provide additional results using i) Kendall Tau and ii) Weighted Kendall Tau correlation coefficients on the Source Architecture Selection experiment.

Table 10: Results on source architecture selection. Shown are Kendall Tau correlation scores (higher the better) computed across all source architectures trained on ImageNet. Results where HASTE modified metrics outperform their baselines are in **bold**.

| Target ($\mathcal{D}_T$) | LEEP | HASTE-LEEP | | NCE | HASTE-NCE | | GBC | HASTE-GBC | |
|---|---|---|---|---|---|---|---|---|---|
| | | CA | CS | | CA | CS | | CA | CS |
| CUB200 | 0.238 | 0.142 | **0.714** | 0.142 | **0.238** | **0.619** | 0.619 | 0.619 | **0.714** |
| StanfordDogs | 0.809 | 0.809 | 0.809 | 0.809 | 0.714 | 0.809 | 0.619 | **0.904** | **0.714** |
| Flowers102 | 0.333 | **0.619** | **0.523** | 0.238 | **0.428** | 0.238 | 0.047 | 0.047 | **0.238** |
| Oxford-IIIT | 0.904 | **1.000** | 0.904 | 0.523 | **0.619** | **0.714** | 0.523 | 0.523 | 0.523 |
| Caltech101 | 0.390 | 0.390 | 0.390 | 0.097 | **0.292** | **0.195** | 0.683 | 0.683 | 0.683 |
| Imagenette | 0.714 | 0.714 | **1.000** | 0.619 | **0.714** | **0.683** | 0.619 | 0.619 | **0.714** |
| PACS-Sketch | 0.000 | **0.097** | **0.097** | 0.000 | **0.097** | **0.195** | 0.487 | 0.487 | **0.585** |

Table 11: Results on source architecture selection task. Shown are Weighted Kendall Tau correlation scores (higher the better) computed across all source architectures trained on ImageNet. Results where HASTE modified metrics outperform their baselines are in **bold**.

| Target ($\mathcal{D}_T$) | LEEP | HASTE-LEEP | | NCE | HASTE-NCE | | GBC | HASTE-GBC | |
| | | CA | CS | | CA | CS | | CA | CS |
|---|---|---|---|---|---|---|---|---|---|
| CUB200 | 0.258 | 0.108 | **0.638** | 0.113 | **0.247** | **0.659** | 0.591 | 0.591 | **0.805** |
| StanfordDogs | 0.865 | 0.865 | 0.865 | 0.865 | 0.672 | 0.865 | 0.746 | **0.952** | **0.805** |
| Flowers102 | 0.376 | **0.705** | **0.644** | 0.389 | **0.611** | **0.400** | -0.119 | -0.119 | **0.031** |
| Oxford-IIIT | 0.925 | **1.000** | 0.925 | 0.587 | **0.678** | **0.721** | 0.692 | 0.530 | 0.530 |
| Caltech101 | 0.535 | 0.535 | 0.535 | 0.345 | **0.482** | 0.238 | 0.723 | 0.723 | 0.723 |
| Imagenette | 0.672 | 0.672 | **1.000** | 0.558 | **0.758** | **0.693** | 0.799 | **0.808** | **0.851** |
| PACS-Sketch | -0.145 | **0.026** | **0.095** | -0.063 | **0.044** | **0.232** | 0.567 | 0.406 | **0.651** |

Table 12: Results on source architecture selection task with LogMe as the baseline. Shown are correlation scores (higher the better) computed across all source architectures trained on ImageNet. Results where HASTE modified metrics perform better than their counterparts are in **bold**.

| Target ($\mathcal{D}_T$) | LogMe | HASTE-LogMe | |
| | | CA | CS |
|---|---|---|---|
| CUB200 | -0.310 | **0.082** | **0.365** |
| StanfordDogs | 0.921 | **0.953** | **0.943** |
| Flowers102 | -0.210 | **0.483** | **0.614** |
| Oxford-IIIT | 0.940 | **0.973** | 0.930 |
| Caltech101 | 0.358 | **0.712** | **0.792** |
| Imagenette | 0.928 | **0.930** | **0.971** |
| PACS-Sketch | -0.423 | **0.677** | **0.117** |

## A.7 RESULTS WITH LOGME BASELINE

We include results on LogME as an additional baseline metric. The results cover two experimental settings - Source Architecture Selection (Table 12), and Target Task Selection (Section 5.2). On average, HASTE-LogME shows an improvement of 120.53% in the source architecture selection experiment, and an improvement of 236.16% in the target task selection experiment.

## A.8 DISCUSSION ON TASK TRANSFERABILITY WORKS

Our work focuses on the problem of estimating transferability of a source model on the target dataset prior to fine-tuning. Formally, this seeks to address two downstream tasks : i) of all the source models, find the most suitable to perform transfer learning on a given target dataset, ii) of all the target datasets, find the most suitable to perform transfer learning on a given source model. Thus, we can note that fine-tuning all the possible models or datasets is not a possible solution here. In stark contrast, some recent transferability works ( Zamir et al. (2018b); Dwivedi & Roig (2019); Song et al. (2020; 2019)) consider models that are pre-trained on one or more tasks, and some further transfer these models to another task, requiring the expensive fine-tuning process. These works only discuss task transferability, i.e., transferring across computer vision tasks such as classification to semantic segmentation, semantic segmentation to depth prediction, etc. More precisely, these works establish task relatedness or how similar is one task to the other and do not propose a transferability metric, which is not the focus of our work. Further, these works are not generalizable as they either perform fine-tuning from scratch or have computational costs similar to fine-tuning, which renders these approaches infeasible for our problem. In addition to this, Zhang et al. (2021a), quantified transferability for the task of Domain Generalization, while Tong et al. (2021) discussed transferability for multi-source transfer, both of which operate in a setting quite different from ours.

## A.9 MOTIVATION BEHIND HARDNESS METRIC

Our approach for measuring hardness is motivated by recent works that have shown that the process of transfer learning shows maximum gains when the images from the source and target tasks are in the similar domains. Further, these recent works measure domain similarity as a function of the distance between source and target samples. In addition to these findings from previous works,

Table 13: Results on target task selection using the fine-tuning method for Caltech101 source models. Shown are correlation scores (higher the better) computed across all target datasets. Results, where HASTE modified metrics perform better than their counterparts, are in **bold**.

| Target ($\mathcal{D}_t$) | LEEP | HASTE-LEEP CA | CS |
|---|---|---|---|
| CUB200 | -0.951 | **0.945** | **0.943** |
| Flowers102 | -0.759 | **0.795** | **0.723** |
| StanfordDogs | -0.887 | **0.847** | **0.842** |
| Oxford-IIIT | -0.899 | **0.852** | **0.476** |
| PACS-Sketch | 0.044 | 0.035 | **0.416** |

our empirical analysis also confirms that hardest samples (i.e., with lowest similarity) obtain lower transfer accuracies than the rest of the samples (as shown in Figure 1), confirming the efficacy of our hardness measure.

## A.10 DEFINITION OF TRANSFERABILITY

Following previous works Bao et al. (2019); Nguyen et al. (2020), we define transferability as to *when a transfer may work, and to what extent*

