# OpenReview forum: "Towards Estimating Transferability using Hard Subsets"
_ICLR.cc/2023/Conference — Submitted to ICLR 2023_

### Official Review · Reviewer_nxzz · 2022-10-23

**Confidence:** 4
**Correctness:** 3
**Technical Novelty And Significance:** 3
**Empirical Novelty And Significance:** 3
**Recommendation:** 5

**Clarity, Quality, Novelty And Reproducibility:**

The paper is mostly clear and quality is good. The novelty of proposed method is moderate. The idea of using hard examples for transferability estimation is interesting. The method seems easy to reproduce.

**Strength And Weaknesses:**

Strength:

1. The paper is clearly written and easy to understand.
2. The idea of using hard examples for transferability estimation is interesting.

Weaknesses:

1. The major concern is the setting of the method. The authors assume all the source samples and target samples are available. However, in practice, in many cases we just have the pre-trained models and a subset of the target samples. This makes the method not applicable in many real-world settings.

2. The method of calculating hardness is kind of ad-hoc. It is more about the similarity to the source data rather than the hardness of the samples.

3. Also, many recent baselines are not considered, such as LogMe (ICML 2021).

**Summary Of The Paper:**

In this paper, the authors propose to identity hard-subset samples in the target dataset for transferability estimation. The main idea is to find the hard examples from the target set by computing the similarities of the source features and target features. The hard examples are used with LEEP, NCE and GBC to improve the performance of transferability estimation of these methods.

**Summary Of The Review:**

The authors propose a method for finding hard examples for transferability estimation. The assumption is however too strong which makes the method not widely applicable.

Post rebuttal:

After reading the rebuttal, I still tend to maintain the score. The main concern is the applicability of the method in a more broad setting for transferability setting. The assumption of having all the samples available is too strong for practical application of the method.

---

> ### Author Response · Authors · 2022-12-11
> **Looking forward to your response**
>
> Dear reviewer,
>
> We sincerely appreciate your valuable comments on our work. In our previous response and the updated manuscript, we have tried our best to address the points raised in your review. Is there any unclear point that we can further clarify?
>
> Thank you again!

---

### Official Review · Reviewer_acad · 2022-10-24

**Confidence:** 2
**Correctness:** 2
**Technical Novelty And Significance:** 3
**Empirical Novelty And Significance:** 3
**Recommendation:** 5

**Clarity, Quality, Novelty And Reproducibility:**

Clarity: the current paper is not quite clearly written (see weaknesses above).
Quality: the experimental section has many datasets, tasks and 3 baseline algorithms. This is quite solid. The theoretical analysis is not so strong.
Novelty: some literature for transferability is missing.
Reproducibility: the paper doesn't attach code in the supplementary.

**Strength And Weaknesses:**

Strengths:
1) The problem of estimating the transferability is interesting and important for transfer learning, and the proposed method is well-motivated.
2) Good literature survey and comparison with three baseline algorithms. The improvement is noticeable.
3) The authors test the proposed transferability metric on a large variety of benchmark datasets. The comparison looks quite comprehensive.
4) Some theoretical analysis is discussed. For example, Lemmas 1 and 2 show that Haste is upper bounded by the average log-likelihood of the hard subset, and lower bounded by the NCE measure plus the log-likelihood of the source label distribution computed over the hard subset.

Weaknesses:
1) What is the definition of transferability? I think even this question is still a hot research topic, e.g., in [1], [2], and these metrics have not been discussed in the paper. The transferability definition is not clearly given either.
2) the activation volume $\mathcal{E}_l$ is not well defined.
3) How would eq. (2) and eq. (4) relate to each other? They use the same notation but seem to be different definitions.
4) In eq. (5), the authors say $\mathcal{T}$ can be any existing transferability metric, but the main goal of the current paper is to propose a new transferability metric. Does that mean HASTE is like a "meta"-transferability metric that can work on top of any existing ones?
5) The connection between the class-agnostic and the class-specific methods are not clearly shown. They seem to be two separate methods.
6) For the theoretical analysis, the discussion of Lemmas 1 and 2 are lacking: why would showing the lower and upper bounds of HASTE be interesting and what is the meaning of it?
7) For the experiments Table 1 and Table 2, the improvement over baselines are quite minor (~0.002 sometimes), while sometimes it helps a lot (like Table 5 and Emotion-IMDB). What is the reason for this? Why for example, does HASTE not improve over GBC at all?


[1] Zhang, Guojun, et al. "Quantifying and improving transferability in domain generalization." Advances in Neural Information Processing Systems 34 (2021): 10957-10970.
[2] Huang, Long-Kai, et al. "Frustratingly easy transferability estimation." International Conference on Machine Learning. PMLR, 2022.

**Summary Of The Paper:**

This paper proposes HASTE for estimating the transferability of a source domain to a target domain. Two techniques are introduced, one is class-agnostic and another is class-specific. The techniques achieve state-of-the-art compared to other concurrent baseline metrics.

--post rebuttal--

Dear authors, thank you for going through my comments and trying to address them. However, I still think the current draft is not ready for submission. As Reviewer bePf also pointed out, the current method (HASTE) is not clearly motivated, and comparison with other transferability metrics is not properly made. Overall, this paper is not clearly written. I would suggest the authors keep polishing the draft for the next round.

**Summary Of The Review:**

In summary, this work proposes an interesting approach using the hard subset to measure the transferability. However, some parts are not clearly written, and the theoretical analysis doesn't fully explain the experiments (it only covers LEEP).

---

### Official Review · Reviewer_bePf · 2022-10-25

**Confidence:** 5
**Correctness:** 3
**Technical Novelty And Significance:** 3
**Empirical Novelty And Significance:** 3
**Recommendation:** 6

**Clarity, Quality, Novelty And Reproducibility:**

The paper is well organized and the proposed method is well provided. However, the motivation of the proposed method is not very strong, and the performance is sensitive to the number of hard samples. Some highly related works and the comparisons with them are missing in the paper, which makes the superiority of the proposed method unclear.

**Strength And Weaknesses:**

Pros:
+ The work paper is easy to follow. The authors clearly present the proposed method and make detailed descriptions about the experiments.

+ The idea of using hard samples to estimate transferability is interesting.

Cons:
My main concerns about the work are two-fold:
- The motivation of the proposed transferability estimation method using hard samples are not well described. The observation that hard examples are more out-of-distribution and challenging than the easy samples in the transfer learning process is not a surprise. However, why using hard samples delivers higher transferability estimation performance, especially provided that the target data is usually filled with both easy and hard samples, or only hard samples (considering the case where source and target domain difference is extremely large), is not explained in the paper. Furthermore, the number of hard samples appears a very important hyperparameter in the proposed method. The ablation study show that the performance of the proposed method is very sensitive to this hyperparameter. How do we set this hyperparameter in our own problem?

- Many highly relevant works are missing from the work, which leaves the superiority of the proposed method unclear. Some are listed as follows:
[1] Deep Model Transferability from Attribution Maps, NeurIPS2019.
[2] DEPARA: Deep attribution graph for deep knowledge transferability, CVPR2020.
[3] Logme: Practical assessment of pre-trained models for transfer learning. ICML2021.
[4] Representation similarity analysis for efficient task taxonomy & transfer learning. CVPR2019.
[5] Taskonomy: Disentangling task transfer learning, CVPR2018.
The authors are encouraged to make discussions on these highly relevant works and clarify the differences between the proposed method and these prior works. Furthermore, any experimental comparisons with these works are especially appreciated.

**Summary Of The Paper:**

The paper proposes to address the problem of estimating transferability from a source to the target domain by using examples from the harder subset of the target dataset. The authors introduce class-agnostic and class-specific techniques to identify harder subsets and show that the proposed method can be used with existing transferability metric to improve their reliability.

**Summary Of The Review:**

The paper is well organized and the proposed method is well provided. However, the motivation of the proposed method is not very strong, and the performance is sensitive to the number of hard samples. Some highly related works and the comparisons with them are missing in the paper, which makes the superiority of the proposed method unclear.

---

### Decision · Program_Chairs · 2023-01-20

**Decision:**

Reject

**Justification For Why Not Higher Score:**

Motivation is not clear. The explanations are not convincing.

**Justification For Why Not Lower Score:**

N/A

**Metareview: Summary, Strengths And Weaknesses:**

In this paper, the authors proposed a new transfer learning method based on hard subsets of target data.

Though the authors provide both theoretical analysis and empirical studies in the paper, the motivation for using the so-called hard subsets of target data to estimate transferability is not clear or convincing. In addition, the calculation of the hardness of hard subsets is ad-hoc. The explanations about why the proposed method is able to achieve superior performance over a number of relevant methods are still not very clear.

In summary, this is a potentially promising work but needs to be further polished for publication.